# Experimental Research on Excitation Condition and Performance of Airflow-Induced Acoustic Piezoelectric Generator

**DOI:** 10.3390/mi11100913

**Published:** 2020-09-30

**Authors:** Zhipeng Li, Jinghao Li, Hejuan Chen

**Affiliations:** School of Mechanical Engineering, Nanjing University of Science and Technology, Nanjing 210094, China; lzp418@njust.edu.cn (Z.L.); Leejh1997@njust.edu.cn (J.L.)

**Keywords:** fuze power supply, airflow-induced acoustic, edge tone, piezoelectric transducer

## Abstract

This paper aims to present a novel airflow-induced acoustic piezoelectric generator that can be used to solve the problem of insufficient power supply of modern intelligent fuzes. The sound waves induced by airflow are the key to power generation performance. It is proposed that an edge tone frequency equal to the acoustic mode frequency is a sufficient condition for evoked acoustic waves, and a design idea and scheme for a universal fuze power supply is provided. We establish the vibration model of the airflow-induced acoustic piezoelectric generator. According to the model, the experimental research on the power generation performance shows that the sound pressure frequency, vibration displacement frequency, and output voltage frequency are consistent. The model provides a design idea for a vibration sensor. At the flow rate of 100.8 m/s, the output power is 45.3 mW, which is much higher than the fuze power sources such as the magnetic backseat generator. Therefore, the airflow-induced piezoelectric generator can effectively solve the problem of the modern fuze less types of power supply and low output energy.

## 1. Introduction

With the development of the modern intelligent fuzes, the expansion of information received and processed by the fuze circuit, detonation control, safety control, and detonation ignition all require the fuze power supply to provide sufficient power, but the problems of an insufficient power supply and fewer types of fuze power supply for the fuze have become increasingly prominent [1]. The airflow-induced acoustic piezoelectric generator is a piezoelectric transducer system that uses the oncoming airflow during the flight of the projectile to induce sound waves, thereby causing piezoelectric vibrations. It is a new type of physical fuze power supply that works in outer ballistics and has the characteristics of good electromagnetic compatibility, a small structure size, and high output power. Therefore, if the drive performance of the airflow-induced acoustic piezoelectric generator can be increased to increase its output power and be used in modern fuzes, it will have high military value and application prospects.

In recent years, scholars from various countries have been seeking new and efficient environmental energy harvesters, and research on piezoelectric energy harvesters based on wind energy has become a current research hotspot. Unstable wind-induced vibration effects mainly include three types: flutter, galloping, and acoustic resonance. Airflow-induced acoustic piezoelectric generators belong to the acoustic resonance type. The flutter-type energy harvester is relatively simple in structure [2,3], energy harvesters of this type have flexible structures [4,5,6,7,8,9,10] and elastically supported rigid body structures [11,12,13,14,15,16], but their vibration model is very complex, and there are still difficulties in the optimization of structural design. Generally, rigid structures have higher energy harvesting efficiency, but larger sizes. The galloping energy harvester is characterized by a large vibration amplitude, a simple aerodynamic model, and its structure is prone to overload. The current study of the galloping type energy collector has made great progress [5], especially in terms of research on aerodynamic forces of different cross-sections [17,18,19,20,21], but there is still a lack of reliable research on galloping energy harvesters. Research on the acoustic resonance energy harvester shows that the vibration signal is stable, the vibration frequency is much higher than the flutter and galloping energy harvesters, the matching resistance is smaller, and the output power is larger. It is mainly suitable for high wind speed environments; for example, it is used in weapon equipment [5,6,7,8,9,10,11,12,13,14,15,16,17,18,19,20,21,22]. The mechanism of acoustic resonance is very complicated, and the establishment of a fluid–structure–acoustic coupling model is difficult. The acoustic structure design mostly relies on empirical equations and is carried out through experimental means. The current research on acoustic resonance energy harvesters is relatively active; Seong-hyok Kim et al. from the United States designed a magnetoelectric generator that is vibrated by airflow [23], and measured the frequency of air vibration through experiments; Lei Junming et al. designed a 34 mm piezoelectric transducer and studied the resonance characteristics of the piezoelectric vibrator of the generator, which proved that the fuze airflow-induced acoustic piezoelectric generator is suitable for low-power fuzes [24]; Xu Wei et al. designed a miniaturized airflow excitation power supply using a micro-electromechanical system [25], and introduced a matching energy harvesting circuit; Zou Huajie et al. focused on the excitation process of air flow generating sound waves, designed an acoustic excitation mechanism of air flow at the annular gap nozzle, and obtained a stable pulsating pressure to excite the piezoelectric transducer structure, thereby generating electrical energy [26,27,28]. However, the above research on the acoustic resonance energy harvester was carried out under a stable excitation source or constant excitation force, and there was no in-depth study of the excitation source performance. Based on the mechanism of airflow-induced acoustic waves, this paper mainly studies the influence and performance of airflow-induced acoustics generator, and provides design parameters and technical support for its application.

This article first introduces the working principle of the airflow-induced acoustic piezoelectric generator, analyzes the conditions of the airflow-induced acoustic waves, establishes the vibration model of the piezoelectric vibrator, and finally conducts experimental tests on the acoustic wave performance and power generation performance.

## 2. Composition and Working Principle of the Airflow-Induced Acoustic Generator System

Figure 1 shows the working principle of the airflow-induced acoustic piezoelectric generator [26,28]. The generator system is composed of an airflow-induced acoustic excitation mechanism and an acoustic–electric transducer. As shown in Figure 1a,b, the airflow-induced acoustic excitation mechanism includes an inflow modulation mechanism (D1) and an acoustic tube (D2), the acoustic–electric transducer (D3) includes a piezoelectric vibrator and an electrical energy processing circuit, and Z is the load on the power supply side.

The structure diagram of the airflow-induced acoustic piezoelectric generator is shown in Figure 2. D1 is composed of an outer inlet and annulus, D2 is composed of a wedge and resonant cavity, and D3 is composed of a piezoelectric vibrator and interface circuit. The windstream enters the annulus through the air inlet and ejects a stable vortex. After the vortex hits the wedge of the resonant cavity, the edge tone is formed. The sound wave encounters the piezoelectric vibrator of the copper substrate at the bottom of the resonant cavity and then forms total reflection, forming a stable standing wave in a short time. The sound intensity in the resonant cavity is amplified, and the stable standing wave in the resonant cavity makes the piezoelectric vibrator perform harmonic vibration like a spring, thereby causing the piezoelectric vibrator to release electric energy.

## 3. Acoustic Frequency Calculation and Power Generation Performance Research

### 3.1. Conditions for Airflow-Induced Sound Waves

When the air flow from the narrow slit is blown to a sharp wedge, a sharp pure tone will occur, which is called the edge tone [29]. The edge tone is actually a solid that faces a high velocity fluid and splits it left and right, causing instability, and then generates periodic vortices, which can be produced in both gases and liquids. When the high-speed jet emitted by the narrow slit flows through the static fluid, due to the contact between the high-speed flow and the static medium on the boundary of the jet, vortices are continuously generated and pushed toward the static fluid. When it reaches the edge, reflection occurs, and the reflected shock wave returns to the nozzle, which stimulates more vortices, and the edge tone produced is very stable.

The empirical equation of edge tone frequency is:(1)$$f_{s}=\frac{v}{4H(1+M_{a})}$$
where $f_{s}$ represents the frequency of the edge tone, *v* represents the velocity of the jet, *H* represents the distance from the nozzle to the wedge, and $M_{a}=\frac{v}{c}$ is the Mach number.

For a small pitch, short resonant cavity acoustic tube, the acoustic modal frequency is calculated based on the propagation of the generated edge tone in the cavity. The sound wave propagates in the cavity, and the disturbance flow field deviates from its average value, which can be expressed as: $p=\bar{p}+\overset{´}{p}$, $\rho =\bar{\rho }+\overset{´}{\rho }$, where *p* represents pressure, *ρ* represents density, $\bar{p}$ represents the Reynolds time-average pressure, $\bar{\rho }$ represents the Reynolds time-average density, $\overset{´}{p}$ represents the pressure oscillation component, and $\overset{´}{\rho }$ represents the density oscillation component. According to the classical acoustic theory [30], the three-dimensional acoustic wave equation can be expressed as:(2)$$\frac{1}{c^{2}}\frac{\partial^{2}\overset{´}{p}}{\partial t^{2}}-\nabla^{2}\overset{´}{p}=0$$

Converting the three-dimensional acoustic wave equation to the one-dimensional standing wave equation gets:(3)$$\{\begin{array}{c} \overset{´}{P}=Ae^{-i\frac{2\pi fx}{c}}+Be^{i\frac{2\pi fx}{c}}\\ \overset{´}{v}=\frac{1}{\rho c}\left(Ae^{-i\frac{2\pi fx}{c}}+Be^{i\frac{2\pi fx}{c}}\right) \end{array}$$

For longitudinal waves, it is necessary to consider the influence of the cavity exit conditions on acoustic characteristics. If we let the sound pressure of *x* = 0 and the velocity of the particle be $p_{1}$ and $v_{1}$, the sound pressure and particle velocity obtained at the end of the acoustic cavity at *x* = *L* are denoted as $p_{2}$ and $v_{2}$, and the natural frequency of the acoustic mode of the resonant cavity can be obtained by the sound wave transmission equation:(4)$$\{\begin{array}{c} p_{2}=p_{1}\cos kL-v_{1}\rho csinkL\\ v_{2}=\frac{p_{1}}{\rho c}sinkL+v_{1}coskL \end{array}$$

By substituting the open end condition $p_{2}$ = 0, the acoustic mode expression in the resonant cavity can be obtained as:(5)$$f=\frac{\left(2k-1\right)c}{4L}\;(k\;=\;1,2,3\cdots )$$
where *k* represents the order of the acoustic mode, and *L* represents the longitudinal length of the cavity.

For the airflow-induced acoustic piezoelectric generator, we put forward a correction plan for *L* through a large number of experiments, and we proved that the length of *L* is not only related to the length of the resonant cavity, but is also related to the diameter of the resonant cavity and the distance from the resonant cavity to the wedge. The revised result is:(6)$$f=\frac{c}{4\left(L+\Delta L+\alpha H\right)}$$
where *f* represents the acoustic modal frequency in the cavity, *c* is the speed of sound, *L* is the length of the resonant cavity, and ∆*L* is the correction parameter; we refer to the correction scheme proposed by [28] where ∆*L* = 0.61*R* and α = 0.4.

It can be seen from Equation (1) that with an increase in flow velocity, the frequency of edge tone production also increases gradually. When the frequency generated by the edge tone is near to the acoustic mode frequency of the resonant cavity, the system will resonate and emit a strong sound. At this time, the vortex–acoustic coupling phenomenon occurs, and the pressure oscillation amplitude increases significantly.

The vorticity acoustic coupling phenomenon is a closed loop system. When the high-speed jet emitted by the narrow slit flows through the static fluid, the boundary of the jet will continuously generate vortices due to the contact between the high-speed flow and the static medium. When a portion of vortexes advance with the jet stream and meet the wedge, more vortexes will be excited and sound will be generated at the same time. When the acoustic signal propagates to the bottom of the resonator, reflection will occur. At this time, the sound signal is fed back to the flow field and has an effect on vortex shedding, and the frequency of the new vortex is adjusted to be consistent with the acoustic modal frequency.

Therefore, the sufficient condition for generating sound waves in the resonant cavity is: $f_{s}\approx f$.

According to the conditions for generating sound waves in the resonant cavity, $f_{s}\approx f$, by substituting Equations (1) and (6) into the equation, the following equation can be obtained:(7)$$\frac{v}{4H(1+M_{a})}\approx \frac{c}{4\left(L+\Delta L+\alpha H\right)}$$

Through simplification, the relationship between structural parameters and flow velocity when sound waves are generated in the resonator must meet the following requirements:(8)$$\frac{H}{L+0.61R+0.4H}\approx \frac{v}{c+v}$$

That is, when the length of the resonance cavity (*L*), the distance from the nozzle to the wedge (*H*), and the diameter of the resonance cavity (*R*), and the flow velocity (*V*) in the generator structural parameters satisfy Equation (8), a stable sound wave can be formed.

### 3.2. Vibration Model of Airflow-Induced Acoustic Piezoelectric Generator

The acoustic–electric conversion model of the airflow-induced acoustic piezoelectric generator, is shown in Figure 3. The mechanical equivalent model is shown in Figure 3a; the piezoelectric vibrator is firmly connected to the equivalent mass (M), and the lower end face is firmly connected to the cover plate. Suppose the mass of M is *m* and the displacement is *x*, it is affected by the external excitation force and internal force, where the internal force refers to the combination of the piezoelectric element’s restoring force, the spring’s resistance and damping force. Then the vibration equation for the piezoelectric vibrator is:(9)$$m\ddot{x}+c\dot{x}+k_{E}x=F-F_{P}$$
where, *F* is the excitation force, that is, the acoustic excitation in the resonant cavity, *F* = *F*_0_ sin*ωt*, *x* is the vibration displacement of the piezoelectric vibrator, *c* is the equivalent damping coefficient of the acoustic–electric transducer, $k_{E}$ is the equivalent stiffness of the mechanical system, and $F_{P}$ is the piezoelectric return force of the acoustic–electric transducer.

The piezoelectric vibrator with a relatively thin circular plate (thickness dimension is much smaller than the radius) is designed to work in a stretching vibration mode parallel to the direction of the electric field, and the piezoelectric vibrator is fixedly installed around it. The electromechanical equivalent model is shown in Figure 3b. The circuit is shown in Figure 3c. The surface area and thickness of the piezoelectric sheet are, respectively, *A* and *b*, and the output voltage and current are, respectively, *V* and *I*. According to the fourth piezoelectric equation shown in reference 9, it is not difficult to obtain the macroscopic relationship between mechanical and electrical quantities as:(10)$$\{\begin{array}{c} F_{p}=F_{T}+F_{K}\\ V=\frac{\alpha b}{A}x-\frac{b^{2}C_{p}}{A^{2}}\int_{0}^{t}Idt \end{array}$$

$k_{\mathrm{PD}}$ is the electrical open-circuit equivalent stiffness of the acoustic–electric transducer, *α* is the piezoelectric stress factor, $C_{p}$ is the static clamping capacitance of the acoustic–electric transducer. $F_{T}$ is the elastic force, and $F_{K}$ is the voltage control force. We can get: $F_{T}=(k_{\mathrm{PD}}-\frac{\alpha^{2}}{C_{p}})x$, $F_{K}=\frac{\alpha A}{bC_{p}}V$. If we suppose $K=k_{E}+2k_{\mathrm{PD}}$, $B=\frac{\alpha A}{bC_{p}}$, then $K=k^{\prime }+\frac{\alpha^{2}}{C_{p}}$. Among them: $k^{\prime }=K_{E}+2K_{\mathrm{PD}}-\frac{\alpha^{2}}{C_{p}}$. Then, Equation (9) is transformed into:(11)$$m\ddot{x}+c\dot{x}+k^{\prime }x+BV=F_{0}\sin\omega t$$

According to the initial conditions of the piezoelectric vibrator vibration, x(0)=0, $\dot{x}\left(0\right)=0$, use $\dot{x}$ to multiply both sides of the Equation (11), and then perform [0, *t*] to find the definite integral to get:(12)$$\frac{1}{2}m{\dot{x}}^{2}+\int_{0}^{t}c{\dot{x}}^{2}dt+\frac{1}{2}k^{\prime }x^{2}+\int_{0}^{t}BV\dot{x}dt=\int_{0}^{t}F_{0}\sin\omega t\dot{x}dt$$

The right side of Equation (12) is equal to the work done by the excitation force *F*; the left side of the equation is equal to the system’s kinetic energy, mechanical loss, elastic potential energy and electric energy generated, so the conservation of energy is satisfied.

The fourth item on the left side of Equation (12) is the electric energy converted by the acoustic–electric transducer, and this is denoted as $W_{D}$. If the clamping capacitor on the power supply side stores electric energy $W_{C}$ and circuit loss $W_{P}$, there should be a relational expression, $W_{D}=W_{C}+W_{P}$, that is:(13)$$\int_{0}^{t}BV\dot{u}dt=\frac{1}{2}C_{p}V^{2}+\int_{0}^{t}VIdt$$
when the system is open, I=0, so $W_{P}=$0, $W_{D}=W_{C}=\frac{1}{2}C_{p}V^{2}$.

If we let $F_{I}=\frac{\alpha b}{A}\int_{0}^{t}Idt$, Equation (11) is transformed into:(14)$$m\ddot{x}+c\dot{x}+Kx=F_{0}\sin\omega t+F_{I}$$

According to the forced vibration theory, the total solution to Equation (14) is: $x=x_{1}+x_{2}$, among them, $x_{1}$ is the general solution, $x_{2}$ is the special solution. They can be expressed as:(15)$$x_{1}=X_{1}e^{-nt}\sin\left(\sqrt{p^{2}-n^{2}}t+\varphi \right)$$
(16)$$x_{2}=X_{2}\sin\left(\omega t+\psi \right)$$

In the equation, $2n=\frac{c}{m}$, $p^{2}=\frac{K}{m}$, $X_{1},X_{2},\varphi$, and ψ are constants, and *p* is the natural frequency of the transducer.

$x_{1}$ is only a transient term, which means that the transient vibration of the system decays exponentially and will disappear soon, at the beginning, the amplitude of the system gradually increases, and when $x_{1}$ disappears, the system will vibrate at a constant amplitude according to $x_{2}$.

So when the system is stable:(17)$$x=x_{2}=X_{2}\sin\left(\omega t+\psi \right)$$

According to Equation (10), voltage is a function of vibration displacement, so voltage and displacement have consistent frequencies. The frequency of displacement in Equation (17) is consistent with the excitation frequency. It can be concluded that the excitation force frequency, vibration displacement frequency, and voltage frequency are consistent.

## 4. Acoustic Performance Experiment

### 4.1. Airflow Induced Acoustic Frequency Experiment

#### 4.1.1. Experimental Prototype Parameters

In accordance with Equation (8), we designed the structure size of the annulus, resonant cavity, and wedge, and selected four experimental prototypes. We took the annulus as *X*, the distance from the nozzle to the wedge as *H*, the length of the cavity as *L*, and the diameter as *D*. The structure parameters are shown in Table 1.

By substituting the mechanical parameters of the four experimental prototypes into Equation (4), the jet velocities at resonance were found to be: $v_{1}\approx 110.27\;m/s$, $v_{2}\approx 90.67\;m/s$, $v_{3}\approx 85.57\;m/s$, $v_{4}\approx 60.28\;m/s$.

In accordance with Equation (14), we took the speed of sound propagation in the air as: c = 340 m/s, and substituted the calculated velocity at resonance of the four experimental prototypes, respectively, and the resonant frequencies of the prototype were found to be the following: $f_{s1}=6.94\;\mathrm{kHz}$, $f_{s2}=5.96\;\mathrm{kHz}$, $f_{s3}=2.85\;\mathrm{kHz}$, and $f_{s4}=2.13\;\mathrm{kHz}$.

Therefore, when the flow rates of the four experimental prototypes are around 110.27 m/s, 90.67 m/s, 85.57 m/s and 60.28 m/s with the same structural parameters, if sound can be emitted and sound frequencies are detected near 6.94 kHz, 5.96 kHz, 2.85 kHz and 2.13 kHz, it can be considered that acoustic waves have been generated in the resonant cavity and resonance has occurred.

#### 4.1.2. Blowing Simulation Test System

Figure 4 and Figure 5 show the blowing test process and experimental system, respectively; we use an air compressor to blow the air to simulate the external ballistic environment. The pulsation pressure sensor measured the sound pressure curve at the bottom of the cavity, and then recorded the measurement value of the pulsation pressure sensor through the data acquisition card, and the laser displacement sensor recorded the vibration displacement curve of the piezoelectric vibrator, and finally through spectrum analysis, analyzed the sound pressure frequency and the vibration frequency of the piezoelectric vibrator. Figure 6 shows a physical picture of the airflow-induced acoustic piezoelectric generator

#### 4.1.3. Sound Pressure Signal Detection

According to the air compressor experimental process, the sound pressure curves of four experimental prototypes under different flow velocities used in the simulation experiment of the air compressor were verified. Figure 7 shows the sound pressure curve at the bottom of the resonant cavity and the spectrum analysis diagram of the sound pressure curve collected by the data acquisition card under the condition of a flow velocity of 127 m/s for the prototype # 1. In Figure 7, V_min_ represents the minimum value, V_max_ represents the maximum value, V_pp_ represents the peak-to-peak value, and F represents the waveform frequency.

According to the sound pressure curve and spectrum analysis diagram shown at the bottom of the resonant cavity in Figure 7, the frequency of the sound pressure vibration was 7.24 kHz. According to the sound pressure curve, the vibration of the sound pressure followed a sinusoidal vibration curve with stable amplitude, which conforms to the assumption of the force given in the acoustic and electrical energy exchange model in Section 3.2.

The sound pressure curves of the four experimental samples in different flow velocity ranges were tested by the same test method and are recorded in Table 2.

Using Table 2, we drew the relationship curve between the sound pressure frequency at the bottom of the resonant cavity and the natural frequency, and fitted the sound pressure frequency curve to calculate the sound pressure frequency value at the resonance flow rate, as shown in Figure 8. $f_{1}–f_{4}$ represent the measured sound pressure frequency at the bottom of the resonant cavity, and $f_{01}$–$f_{04}$ represent the natural frequency of transducers 1–4.

In accordance with the fitting curve in Figure 8, the resonance flow rates of the four prototypes were substituted into their fitting curve equations. The sound pressure frequency of prototype #1 at its resonance velocity $v_{1}=110.27\;m/s$ was $f_{1}=6.929\;\mathrm{kHz}$, the sound pressure frequency of prototype #2 at its resonance velocity $v_{2}=90.67\;m/s$ was $f_{2}=6.022\;\mathrm{kHz}$, the sound pressure frequency of prototype #3 at its resonance velocity $v_{3}=85.57\;m/s$ was $f_{3}=2.918\;\mathrm{kHz}$, and the sound pressure frequency of prototype #4 at its resonance velocity $v_{4}=60.28\;m/s$ was $f_{4}=2.086\;\mathrm{kHz}$. It can be seen from the fitting equation that the slope of the curve gradually decreased, indicating that the longer the resonant cavity, the closer the detected sound pressure frequency value is to the first-order resonance frequency, which verifies the rationality of the structural parameter design presented in Section 3.1.

It can be seen from the table that the prototype did not produce acoustic resonance in all flow velocity ranges. Prototype #1 could only generate sound waves when the flow velocity reached 64 m/s or more, and no sound waves were generated below 64 m/s. When prototype #3 was at 53 m/s, the sound pressure curve could not be detected at the bottom of the resonance cavity. Prototypes #2 and #4 were shown to emit sound at the flow velocities of 53–159 m/s, and a sound pressure curve was detected at the bottom of the resonator cavity. However, for flow velocities higher than 159 m/s, further verification is needed to determine whether the sound phenomenon will be broken. A limitation of the experimental conditions was that the flow velocity could not reach more than 159 m/s; this is an area for future research.

Table 3 shows the actual sound pressure frequency ($f_{1}$), the theoretical resonance frequency ($f_{2}$), absolute error (∆) and relative error (δ) of the four experimental prototypes near the calculated flow velocity (V).

It can be seen from Table 3 that near the calculated sound velocity, all the four experimental prototypes were able to emit sound, and a stable sound pressure waveform was detected at the bottom of the resonant cavity. The relative deviation was within 3%, which is considered consistent with the theoretical calculation shown in Section 3.1.

### 4.2. Experimental Analysis of Power Generation Performance

#### 4.2.1. Experimental Verification of the Frequency Characteristics

We used prototype #2 to conduct an experimental study of the generation of electric energy. We took three piezoelectric vibrators with similar natural frequencies and installed them on the bottom of the resonant cavity. We used the data acquisition card to collect the open-circuit voltage output of the piezoelectric vibrator, and we used the laser displacement sensor to measure the vibration displacement of the piezoelectric vibrator. Figure 9 and Figure 10 show the voltage curve and displacement curve of the piezoelectric vibrator no. 1 at a flow rate of 100.8 m/s, as well as its spectrum analysis diagram.

For the open-circuit voltage curve of the piezoelectric vibrator, through a spectrum analysis, it was found that the open-circuit voltage frequency was: $f_{V}=5.85\;\mathrm{kHz}$. The main peak of the displacement vibration frequency of the piezoelectric vibrator was read through the frequency spectrum analysis chart of the displacement curve: $f_{X}=5.86\;\mathrm{kHz}$. The error between the output open-circuit voltage frequency $f_{V}$ and the vibration displacement frequency $f_{X}$ of the piezoelectric vibrator was only 10 Hz, and the relative error was only 0.17%. It can be considered that the two frequencies were consistent. It was shown in Section 3.1 that the sound pressure frequency of prototype #2 at a flow rate of 101 m/s was $f_{s}=6.11\;\mathrm{kHz}$. The error between the sound pressure frequency f and the vibration displacement frequency $f_{X}$ of the piezoelectric vibrator was 250 Hz, and the relative error was 4.2%. Therefore, it can be considered that the three have consistent frequencies and conform to the calculation results given in Section 3.2.

#### 4.2.2. Generated Power Conversion Efficiency

We connected a 6 kΩ resistor to the output end of the airflow-induced acoustic piezoelectric generator, and recorded the output power of the resistor and the voltage value when the system is open, as shown in Table 4. *u* represents the peak-to-peak value of the open-circuit voltage of the generator, $f_{V}$ represents the frequency of the open-circuit voltage, and *P* represents the output power when the external load of the airflow-induced acoustic piezoelectric generator was 6 kΩ.

The calculation of the airflow power can be transformed by the kinetic energy equation, into:(18)$$E=\frac{1}{2}mv^{2}=\frac{1}{2}\rho S\Delta tv^{3}$$
where *E* is kinetic energy, *ρ* is the air density, *v* is the jet velocity, ∆*t* is the observation time, *S* is nozzle area, and *m* is the mass.

Through Equation (18), the expression of airflow power can be obtained as:(19)$$P_{a}=\frac{1}{2}\rho S\Delta v^{3}$$

Therefore, the energy conversion efficiency of the airflow-induced acoustic piezoelectric generator can be expressed as
(20)$$\eta =\frac{P}{P_{a}}$$

According to the data presented in Table 4, the maximum energy conversion efficiency of the airflow-induced acoustic piezoelectric generator reached its maximum when the flow velocity was v = 79.79 m/s, and the maximum energy conversion efficiency was *η* = 4.37‰. At the flow rate of 100.8 m/s, the maximum output power reached its maximum value, *P*_max_ = 5.3 mW, while the average output power of the fuze magnetic rear seat generator was only 0.18 mW [31]. Therefore, the power of the airflow-induced acoustic piezoelectric generator is much higher than the fuze magnetic rear seat generator. The current rockets projectiles fuze power supply voltage requirement is greater than 6 V, and the small missile fuze power supply voltage requirement is greater than 12 V [32,33]. According to Table 4, 37.256 V <u <61.696 Vso the effective value of the output voltage is $13.172\;V\;<u_{0}\;<21.813\;V$; this is enough to meet the requirements for the use of rocket projectiles and small missile fuze power.

## 5. Conclusions

The structural design of the airflow-induced acoustic piezoelectric generator must coordinate with the mechanism of airflow-induced acoustic waves. The condition for airflow-induced acoustic waves is that the edge tone frequency must be equal to the acoustic mode frequency of the resonant cavity. The results of the blowing experiment and parameter test of the airflow-induced acoustic piezoelectric generator, show that the acoustic signal in the resonant cavity is a sinusoidal signal with stable frequency, and the acoustic wave is consistent with the frequency of the piezoelectric vibrator displacement and output voltage. The output power reaches tens of milliwatts, which is enough to meet the requirements for the use of fuze power supplies for rocket projectiles and small missiles, and can effectively solve the problems of low modern fuze power supplies and low output energy.

The structural design scheme of the airflow-induced acoustic piezoelectric generator may be helpful in developing a universal design of the fuze power supply. The three frequency characteristics provide a design idea for a vibration sensor, which can express high-frequency vibration signals that are difficult to directly measure by installing a piezoelectric vibrator and testing the frequency of the open-circuit voltage.

## Figures and Tables

**Figure 1 micromachines-11-00913-f001:**
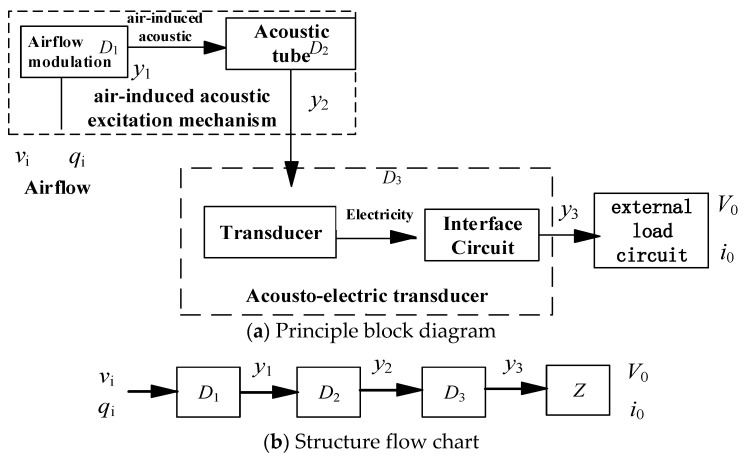
Working principle of the airflow-induced acoustic piezoelectric generator.

**Figure 2 micromachines-11-00913-f002:**
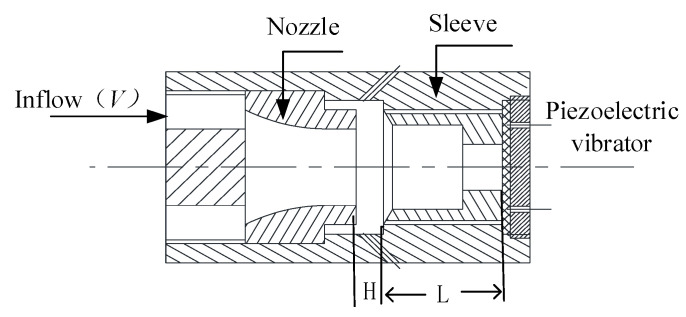
Structure diagram of the airflow-induced acoustic piezoelectric generator.

**Figure 3 micromachines-11-00913-f003:**
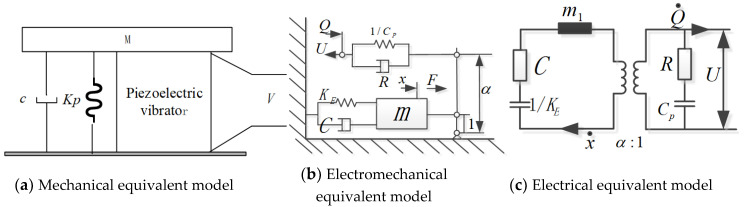
Equivalent model of the airflow-induced acoustic piezoelectric generator.

**Figure 4 micromachines-11-00913-f004:**
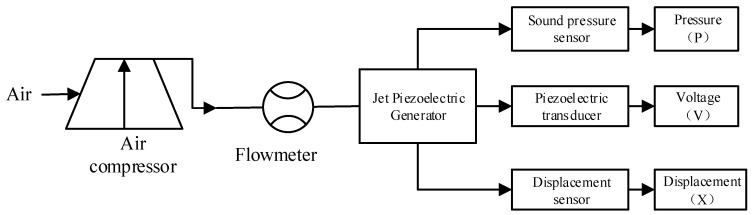
Blowing simulation test process.

**Figure 5 micromachines-11-00913-f005:**
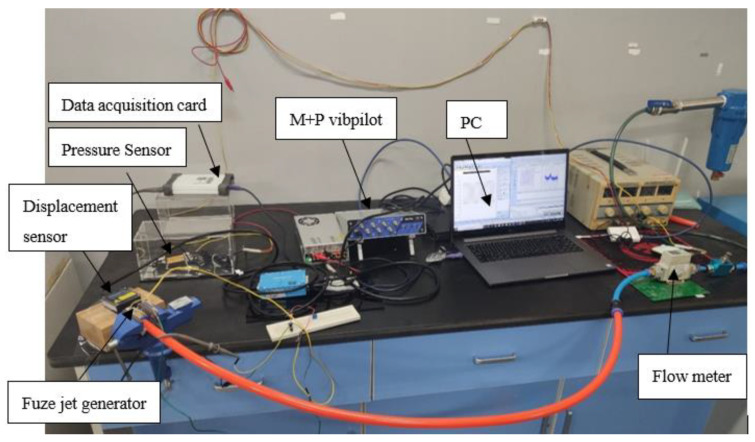
Blowing experimental test system.

**Figure 6 micromachines-11-00913-f006:**
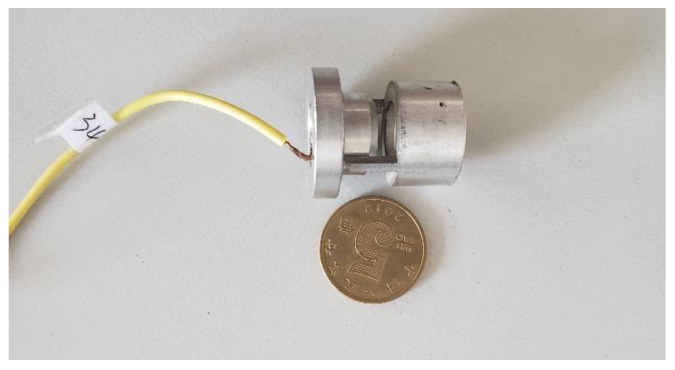
Physical picture of prototype #1.

**Figure 7 micromachines-11-00913-f007:**
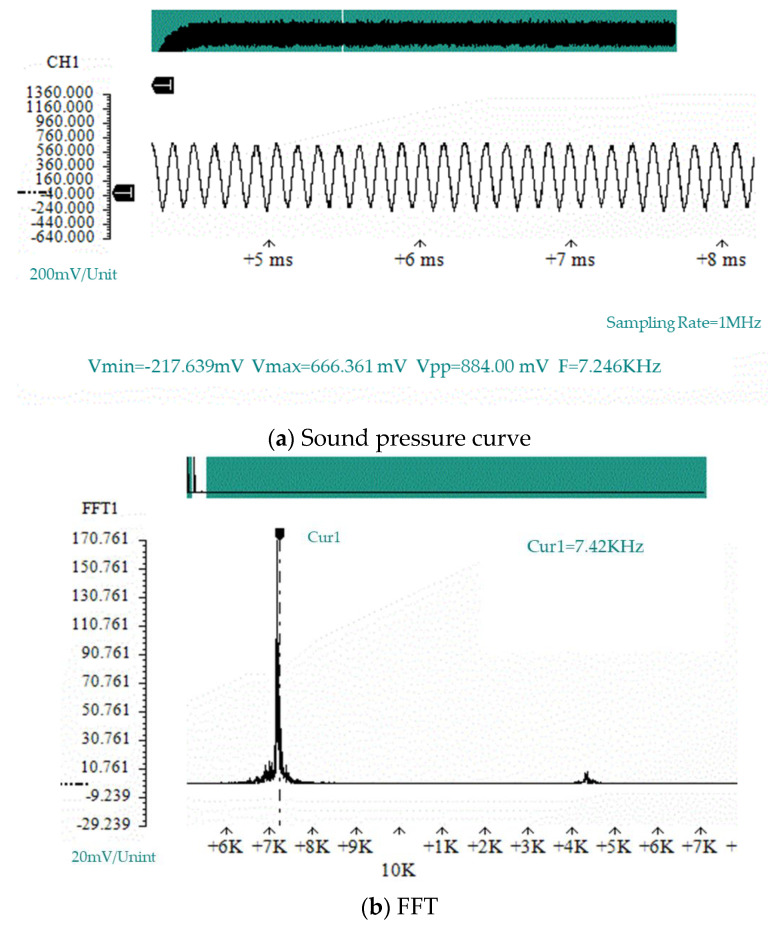
The sound pressure at the bottom of the resonant cavity of prototype #1 when V = 127 m/s.

**Figure 8 micromachines-11-00913-f008:**
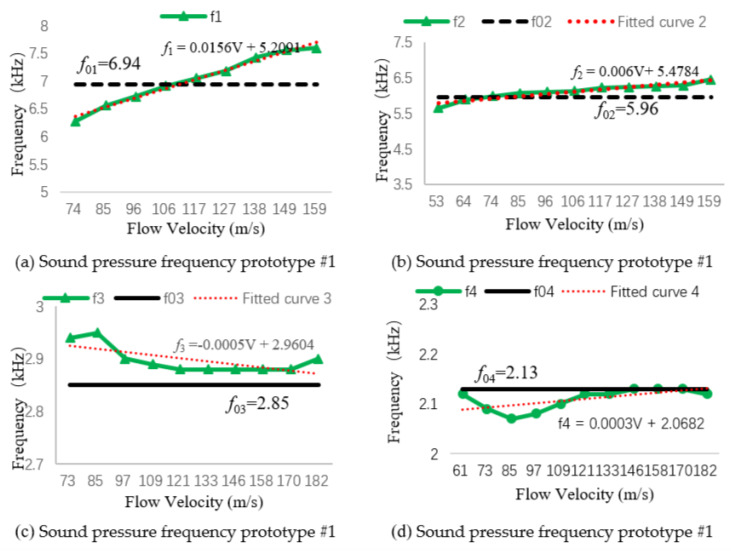
Sound pressure frequency and first-order resonant frequency.

**Figure 9 micromachines-11-00913-f009:**
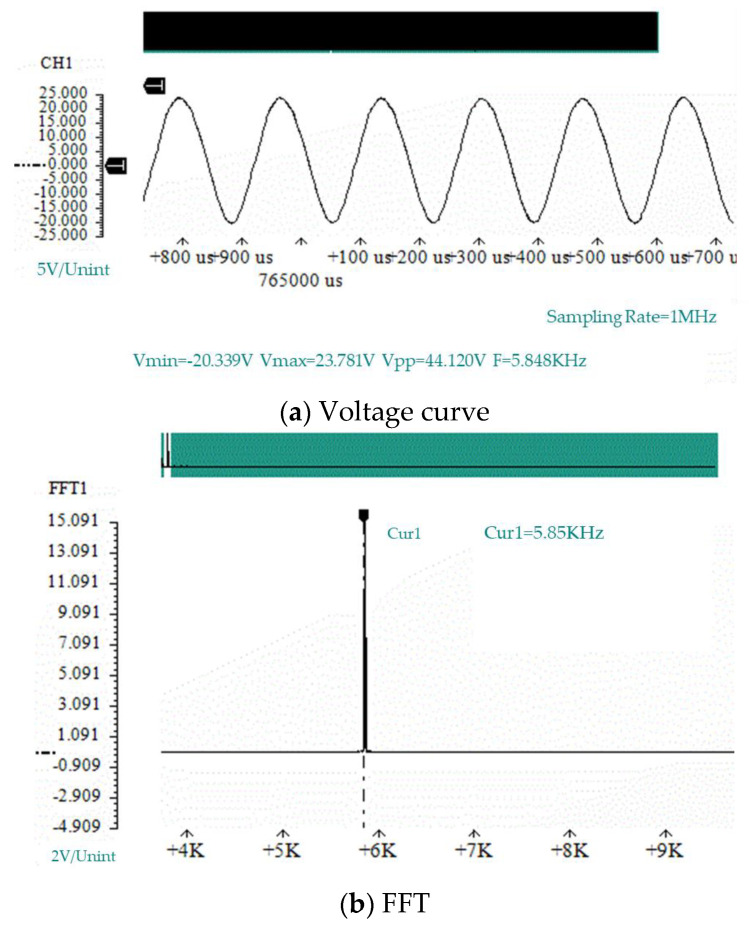
The open-circuit voltage of piezoelectric vibrator no. 1 of prototype #2 at V = 100.8 m/s.

**Figure 10 micromachines-11-00913-f010:**
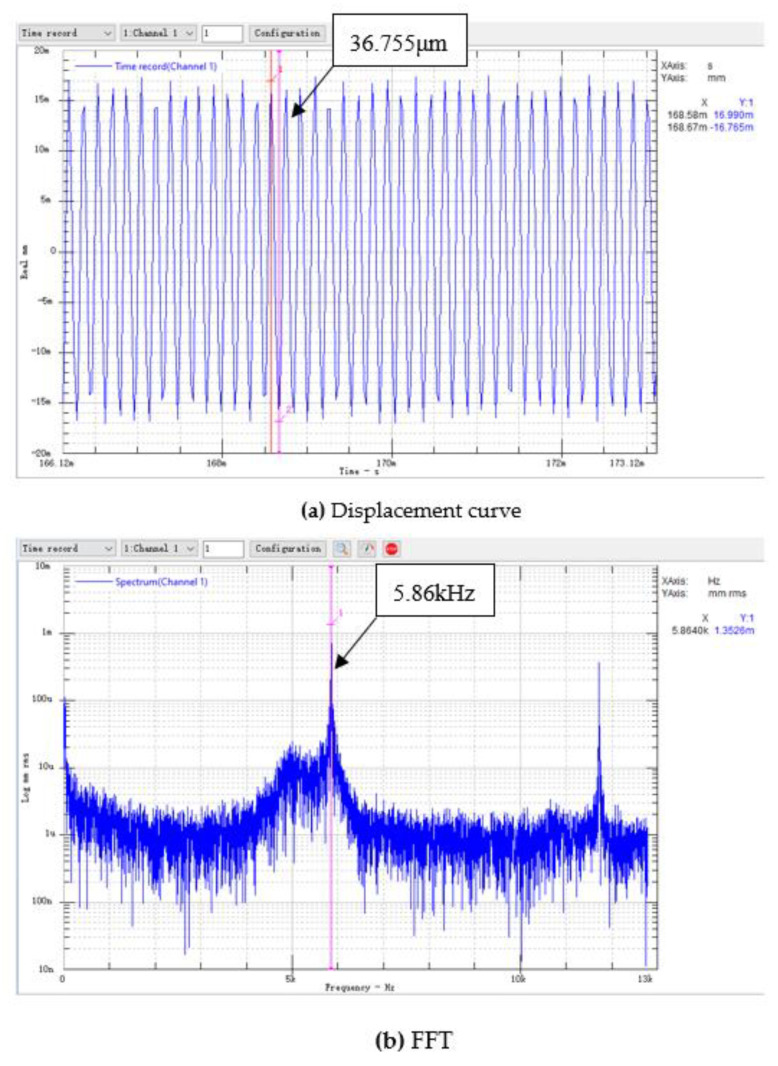
The displacement curve of piezoelectric vibrator no. 1 of prototype #2 at V = 100.8 m/s.

**Table 1 micromachines-11-00913-t001:** Structural parameters of the experimental prototype.

Experimental Prototype	*L* (mm)	*H* (mm)	*X* (mm)	*D* (mm)
1#	8	3	1	10
2#	10	3	1	10
3#	25	6	2.5	8
4#	35	6	2.5	8

**Table 2 micromachines-11-00913-t002:** Sound pressure parameters at the bottom of the resonant cavity.

Flow Velocity (m/s)	1#Frequency (kHz)	2#Frequency (kHz)	3# Frequency (kHz)	4# Frequency (kHz)
53	0	5.64	0	2.12
64	0	5.88	2.94	2.09
74	6.27	5.98	2.95	2.07
85	6.56	6.06	2.9	2.08
96	6.72	6.1	2.89	2.1
106	6.91	6.12	2.88	2.12
117	7.05	6.22	2.88	2.12
127	7.24	6.23	2.88	2.13
138	7.42	6.26	2.88	2.13
149	7.55	6.28	2.88	2.13
159	7.59	6.452	2.9	2.12

**Table 3 micromachines-11-00913-t003:** Actual and theoretical resonance frequency values.

Experimental Prototype	V (m/s)	$f_{1}\;(\mathrm{kHz})$	$f_{2}\;(\mathrm{kHz})$	Δ (kHz)	δ (%)
1#	110.27	6.929	6.94	0.011	0.16
2#	90.67	6.022	5.96	0.062	1.04
3#	85.57	2.918	2.85	0.068	2.38
4#	60.28	2.086	2.13	0.044	2.07

**Table 4 micromachines-11-00913-t004:** Output test data for the generator.

V (m/s)	*u* (v)	$f_{V}$	*p* (mW)
58.51	37.256	5682	18.1
69.15	52.246	5714	35.7
79.79	61.696	5747	45.1
90.43	56.855	5780	42.6
100.85	58.443	5814	45.3

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
