# Peer review of "Experimental Research on Excitation Condition and Performance of Airflow-Induced Acoustic Piezoelectric Generator"

_micromachines, 2020, doi:10.3390/mi11100913_

Round 1

Reviewer 1 Report

The article submitted for evaluation under the title: Experimental Research on Excitation Condition and Performance of Airflow-induced Acoustic Piezoelectric Generator attempts to design an innovative acoustic airflow-induced piezoelectric generator. First, the author presents the theoretical foundations of the phenomenon as well as the main goals and assumptions of the project. In the following chapters, the electromechanical model of the generator was presented and preliminary tests were carried out.
Analyzing the test results, it was found that the assumed goals were achieved, and the output power reaches tens of milliwatts, which is sufficient to meet the requirements of using fuse feeders, e.g. for small missiles.

The article contains numerous stylistic and editing errors that should be corrected before printing, such as:

1) In Equation 4, the parenthesis is redundant.

2) On page 2, there seems to be no need for the period in the phrase “Figure. 1 "

3) It is also worth making the case of letters uniform; once the author writes on page 11 as "Figure 9 and Figure 10", while on page 7 he writes as "Figure 4and figure 5"

4) On page 4 I found the following entry: "According to the conditions for generating sound waves in the resonant cavity: ??≈? , By substituting equations 6and 1 into the equation, the following formula can be obtained". It seems that "By" is part of the previous sentence, so either a period is missing or “By” should be written in lowercase.

5) The quality of the photo number 7 is not the best, I understand that it is a photo taken directly from the display of the measuring apparatus.

6) Please consider putting the table number 3 on one page, which will increase its readability.

It is recommended that the author autocorrect the article with particular emphasis on editorial error.

Author Response

Dear Reviewer:

According to your valuable suggestions, modifications one-by-one are as the follows:

  1. Calculate formula 4 further, remove the parentheses.
  2. Deleted the word " Figure. 1" in 1 paragraph of 2 section on 2 page.
  3. The case of letters in the article has been changed to be consistent.
  4. “By” should be written in lowercase, it has been changed in the manuscript.
  5. Some pictures are obtained by taking screenshots directly on the test software. I have tried to improve the quality of the pictures by other means. These changes have been shown on the manuscript.
  6. I have put Table 3 on one page, and all the changes in the article can be displayed on the manuscript

Thank you and best regards.

Yours sincerely

Reviewer 2 Report

The manuscript analyses the ‘Airflow-induced Acoustic Piezoelectric Generator’. The manuscript consists of 10 figures and 4 tables. The topic is very interesting for space applications. The subject may attract interest to the readers, nevertheless, the reviewer has several comments and concerns regarding the content and presentation.

Remark: The manuscript should have the line number.

Remark:  Please rewrite the following sentences for better understanding.

  • Abstract, Line no. 5-8: By establishing a vibration ………… for a vibration sensor.
  • Abstract, Line no. 8-10: When the flow rate is………………. fuze power sources.
  • Page 5: The second sentence of section 2.2 is too large. It is difficult to read.
  • Page 6: The sentence after Eq. (17) is difficult to read, please rewrite it. “According to Equation 10, .........frequency are consistent.”

Remark: Authors claimed that the maximum output power is 45.3mW at a flow rate of v=100.85m/s. I suggest authors to compare these results with previous studies. Authors also claimed that this power is enough to meet the requirements for the use of rockets and small missile fuze power. Please mention the source of this information and cite appropriate references.

Remark: The manuscript should be edited extensively before publication. Some of the sentences are written very clumsily. In many sentences, authors used capital letters after the comma (,). Such errors are not acceptable in scientific writing.

Remark: Please improve the figure quality.

Remark: Please rewrite the introduction. Please explain the novelty of the work in abstract and conclusion.

Remark: Author should critically and quantitatively compare the proposed technique with other reported techniques in this area.

Remark: References are not in MDPI style. Please correct it.

Author Response

Dear Reviewer:

According to your valuable suggestions , especially the suggestion to add " the source of this information and cite appropriate references about the requirements for the use of rockets and small missile fuze power", which increases the completeness of the article, thank you again. Modifications one-by-one are as the follows:

  1. Line numbers have been added to the manuscript.
  2. Rewrite lines 5-8 in the abstract: By establishing a vibration ………… for a vibration sensor.

        Rewrite lines 8-10 in the abstract: When the flow rate is……………….          fuze power sources.

        Rewrite the second sentence in section 2.2 on page 5.

        Rewrite the second sentence after Eq. (17) on page 6.

  1. References on the output requirements of rocket fuze power supply and small missile fuze power supply have been added.
  2. Some editing errors are rechecked in the manuscript.
  3. Some pictures are obtained by taking screenshots directly on the test software. I have tried to improve the quality of the pictures by other means. These changes have been shown on the manuscript.
  4. The main content of the article not prominent as the introduction and principle are in one part in my previous manuscript. I have divided it into two parts.
  5. The second paragraph of the introduction introduces the current research status of piezoelectric energy harvesters based on wind energy.
  6. The format of references has been corrected.

Thank you and best regards.

Yours sincerely

Round 2

Reviewer 1 Report

The author corrected the indicated editing errors and the reviewer's suggestions.

Reviewer 2 Report

Authors have revised the manuscript as per the reviewer's comments. The manuscript can be accepted for publication. However, the manuscript contains several long sentences (more than 3 lines).

I suggest authors to take care of this issue. I also recommend authors to use professional English editing services. Some of the sentences are not grammatically correct as per scientific writing.